# mimic-one: a Scalable Model Recipe for General Purpose Robot Dexterity

**Abstract:** We present a diffusion-based model recipe for real-world control of a highly dexterous humanoid robotic hand, designed for sample-efficient learning and smooth fine-motor action inference. Our system features a newly designed 16-DoF tendon-driven hand, equipped with wide angle wrist cameras and mounted on a Franka Emika Panda arm. We develop a versatile teleoperation pipeline and data collection protocol using both glove-based and VR interfaces, enabling high-quality data collection across diverse tasks such as pick and place, item sorting and assembly insertion. Leveraging high-frequency generative control, we train end-to-end policies from raw sensory inputs, enabling smooth, self-correcting motions in complex manipulation scenarios. Real-world evaluations demonstrate up to 93.3% out of distribution success rates, with up to a +33.3% performance boost due to emergent self-correcting behaviors, while also revealing scaling trends in policy performance. Our results advance the state-of-the-art in dexterous robotic manipulation through a fully integrated, practical approach to hardware, learning, and real-world deployment.

**Keywords:** Dexterity, Manipulation, Self-Correction

## 1 Introduction

Robotic dexterity—the ability to manipulate a wide range of objects with precision, speed, and adaptability—remains one of the grand challenges in robotics. Although recent advances in learning-based control, model architectures, and robotic hardware have enabled significant progress in domains like locomotion and grasping, achieving general-purpose, real-world dexterous manipulation still demands a holistic integration of perception, data, hardware, and control. Human hands, through evolution, have become masterful tools for contact-rich manipulation. Can we equip robots with similarly versatile capabilities?

In this work, we introduce **mimic-one**, a scalable recipe for real-world, general-purpose robotic dexterity based on high-frequency generative models trained end-to-end with imitation learning. Our system centers around a newly developed 16 degree-of-freedom (DoF) humanoid robotic hand, capable of fine-grained, tendon-driven control. The hand is mounted on a 7-DoF Franka Emika Panda arm and equipped with wide-FOV wrist cameras to provide rich, low-latency feedback from diverse viewpoints. The entire system is optimized for smooth inference and sample efficiency and is powered by a diffusion-based control policy trained from raw image and proprioceptive observations.

To collect diverse high-quality demonstrations, we design an extensible teleoperation system that supports both glove-based and Apple Vision Pro-based control interfaces. These enable rapid data acquisition across a variety of real-world tasks—including pick-and-place of deformable objects, precise insertions, and complex object sorting—all selected to challenge the limits of contemporary manipulation systems. Crucially, our methodology includes a robust data collection protocol to identify and mitigate failure modes through targeted self-correction data, which we show is vital for improving robustness and policy generalization.

Submitted to the 9th Conference on Robot Learning (CoRL 2025). Do not distribute.

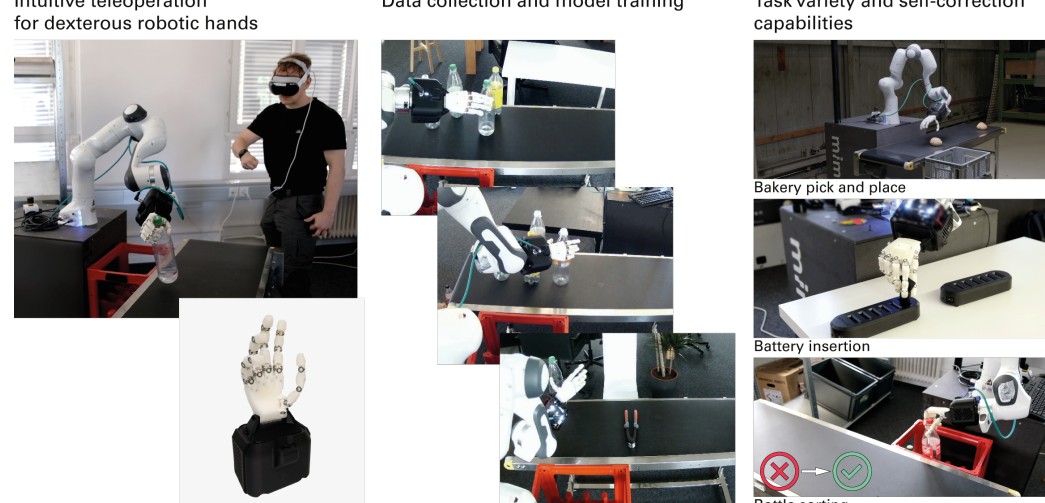

Intuitive teleoperation for dexterous robotic hands

Data collection and model training

Task variety and self-correction capabilities

Bakery pick and place

Battery insertion

Bottle sorting

Figure 1: **mimic-one** is a scalable model recipe and data collection protocol for general purpose dexterous manipulation. Our system features a newly designed 16-DoF tendon-driven hand, a 7-DoF Franka Emika Panda robot arm, and a VR teleoperation system for data collection. We showcase the framework across difficult, dynamic real-world manipulation scenarios, which highlight the system's high frequency fine-motor skills, visual generalization, and ability to recover from failures.

We demonstrate that mimic-one policies are not only capable of completing these tasks with high success rates, but also natively exhibit recovery from failed grasps and adaptive corrections. Our ablation studies reveal key data curation and representation decisions that dramatically impact performance. Importantly, we uncover scaling trends that highlight how policy generalization and reliability improve downstream of data diversity and curation.

## 2 Methodology

### 2.1 The mimic dexterous robotic hand

With the ultimate goal of demonstrating highly dexterous, human-like manipulation, we developed a novel "mimic" dexterous robotic hand (Fig. 4.c). The hand features 20 joints with 16 DoF, actuated by an antagonistic tendon-driven mechanism with slack compensation. The rigid articulated bodies are padded with a soft silicone skin at relevant contact surfaces. The system successfully demonstrated lifting capacity for payloads exceeding 7 kg with 5-fingered power grasps. The hand also features two wide-angle RGB wrist cameras (Fig. 4.b), with the aim of providing rich, low-latency feedback from diverse viewpoints. An emphasis is placed on abduction and adduction for all fingers and the anatomically accurate opposability of the thumb. This design choice allows an intuitive user experience for teleoperation, narrowing the embodiment gap in between the robot and the human operator. Other, less relevant DoF of human biomechanics are simplified with joint coupling and underactuated mechanisms to reduce system complexity and weight.

### 2.2 Robot setup and teleoperation

Our robot station setup consists of a Franka Emika Panda robot arm, a mounted mimic dexterous hand, and one additional external "workspace" RGB camera (Fig. 4.b). Demonstration data is collected via teleoperation using one of two methods: 1. Manus gloves for hand motion capture and SpaceMouse control for wrist poses; or 2. Apple Vision Pro hand and wrist tracking (Fig. 4.a). The Franka Emika Panda robot arm runs a low-level Cartesian impedance controller from [1], following a high-level target pose commanded by the teleoperator. We apply a retargeting algorithm to the hand finger joints, as detailed in App. B.1.

## 2.3 Model recipe and data collection protocol

Our recipe relies on several key design choices for state and action representation that significantly boost performance and generalization for dexterous policies. These design choices mainly involve: **1.** Using **cartesian end effector poses** as states and actions; **2.** Using a **"relative" end effector state and action representation**; **3.** Using **absolute hand joint angles** state and actions; **4.** Using **6D rotations**. We detail the model recipe choices in further detail in App. C.1.

At the same time, we observe that, irrespective of scale, obtaining high success rates with imitation learning policies is facilitated when following a specific **data collection protocol**, involving: **1. Data collection** with randomized object positions, distractors and backgrounds; **2. Data labeling** for success/failure; **3. Data curation**, filtering for high-quality trajectories; **4. Policy training**; **5.** Iterative **self-correction data collection** for the common policy failure modes. We further detail the protocol in App. C.2.

## 2.4 Policy architecture

Our goal is to train a policy $\pi(a_{t:t+H_a}|o_{t-H_o:t})$ that maps from a sequence of observations $o_{t-H_o:t}$ to future actions $a_{t:t+H_a}$ for the dexterous hand plus arm. Our observations $o_i = (l_i, I_i)$ are composed of low-dimensional sensory readings $l_i$, such as proprioceptive robot joint angles and end effector poses, together with RGB images $I_i$ from the cameras mounted on the workstation and the wrist. We predict an entire "action chunk" of length $H_a$, as this makes it easier to learn and generate smooth, high-frequency motions that are temporally consistent and exhibit lower compounding error.

Our chosen approach for simple but effective end-to-end high-frequency generative control is to use a UNet Diffusion Policy architecture [2], receiving as conditioning input an observation horizon with three RGB images from the wrist-mounted and overhead cameras and proprioceptive state inputs (end-effector pose, joint angles). The low-dimensional proprioceptive state inputs are projected to the embedding dimension with a linear layer, while the RGB images are fed into CLIP [3] pre-trained ViT-B/16 [4] encoders with duplicated weights for each camera stream. The embeddings are concatenated and passed to the UNet Diffusion Policy backbone, which samples the action chunks by denoising a vector of sampled Gaussian noise (Fig. 5.c).

# 3 Results

We evaluate our model recipe on three challenging dexterous manipulation benchmarks designed to test generalization, precision, and self-correction capabilities (Fig. 6). These tasks intentionally move beyond simple pick-and-place scenarios to probe the limits of dexterous control.

**General pick-and-place (Bread Picking):** This task involves picking deformable bread loaves from varied surfaces (static table, moving conveyor belt) and placing them into a box. Key challenges include handling the deformable object gently and adapting to randomized loaf positions and diverse surfaces (varied table colors, conveyor). **Evaluation:** Assessed on unseen table/background settings with arbitrary loaf starting positions. Success requires the loaf to be gently placed into the target container, with the policy allowed to self-correct errors.

**Complex placement (Bottle Sorting):** This task mimics a recycling scenario, requiring the robot to grasp plastic bottles (potentially slippery, empty or partially filled) sideways on a conveyor belt and insert them precisely into a bottle rack with variable slot occupancy. Challenges include achieving a stable sideways grasp and accurate placement into a constrained slot. **Evaluation:** Performed on the training conveyor belt but with unseen background settings, arbitrary bottle positions, and randomized rack occupancy. Success requires placing a bottle into an empty slot, permitting recovery from initial misplacements (e.g., re-orienting a poorly slotted bottle).

**Precise insertion (Battery Insertion):** This task requires high precision, involving picking a battery from one rack, transporting it, and inserting it fully into a specific slot in a second rack, including a final "punch" motion to ensure it's connected. The main challenges lie in the precise alignment

needed for insertion (which depends on the battery's pose in hand, not just the end-effector pose) and executing the dynamic push. **Evaluation:** Conducted on an unseen table/background with randomized rack positions. Success requires the battery to be picked, correctly inserted into the target slot, and fully "punched" in.

**Quantitative evaluation.** We assessed policy performance by varying the amount of training data (Fig. 2). Success rates improved significantly with more data across all tasks: Bread Pick (22.5%@20% data to 93.3%@100%), Bottle Sort (25.0%@50% to 75.0%@100%), and Battery Insertion (12.5%@50% to 37.5%@100%). These results, achieved with relatively modest dataset sizes, highlight the effectiveness of our data collection protocol.

Crucially, incorporating self-correction trajectories markedly improved robustness (Fig. 2, dashed vs. solid bars). Success rates increased substantially when allowing for recovery behaviors: Bread Pick (+26.6%), Bottle Sort (+33.3%), and Battery Insertion (+25.0%), demonstrating the value of targeted failure recovery data.

**Ablation study.** We validated key components of the **mimic-one** recipe via ablations on the Bread Pick-and-Place task (Fig. 3). Using suboptimal action representations (**i**, **ii**), limiting data diversity (**iii**), omitting data filtering (**iv**), or excluding self-correction data (**v**) all led to significantly lower success rates (ranging from 5.0% to 56.7%) compared to the full recipe (**vi**, 93.3% success rate). This confirms the importance of each element in our proposed methodology, particularly the relative action representation with the correct base frame, data diversity/curation, and inclusion of self-correction trajectories.

# 4 Conclusion

We presented **mimic-one**, a scalable recipe for achieving general-purpose robotic dexterity using a novel 16-DoF humanoid hand and a diffusion-based generative control policy. Our approach leverages efficient teleoperation for data collection, incorporating a crucial protocol for targeted self-correction data, and relies on specific design choices like relative Cartesian action representations for effective end-to-end imitation learning. Experiments demonstrated the system's capability on challenging real-world tasks, showcasing high success rates, smooth control, generalization, and emerging self-corrective behaviors. Ablations and scaling results validated our methodology's key components. The mimic-one framework offers a practical and reproducible path towards bridging the gap between generative models and the demands of robust, real-world dexterous manipulation.

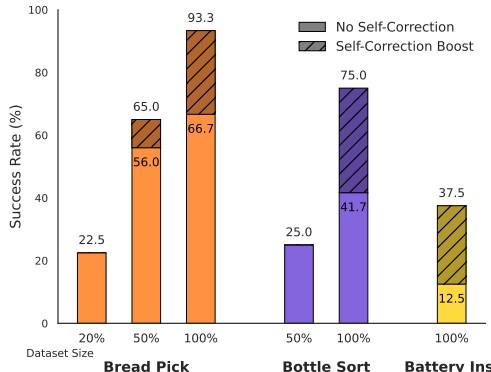

Figure 2: Task success rates vs. dataset scale. Dashed bars include self-corrected successes; solid bars count any error as failure.

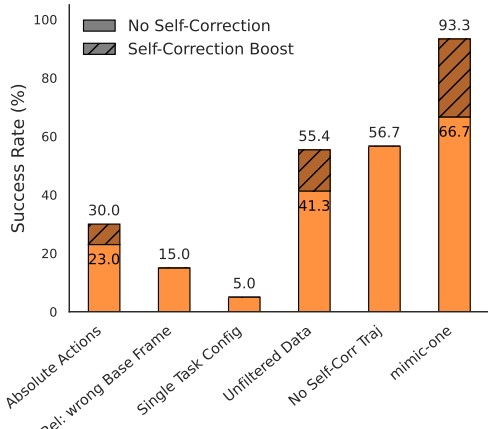

Figure 3: Ablation study on Bread Pick-and-Place, comparing the full **mimic-one** recipe (**vi**) against variants removing key components: (**i**) Absolute actions, (**ii**) Incorrect relative base frame, (**iii**) Single task configuration, (**iv**) Unfiltered data, and (**v**) No self-correction data.

# Appendix

## A Related Work

**Dexterous manipulation hardware.** Robotic hand design has progressed toward replicating human-level dexterity via either high-fidelity anthropomorphic mechanisms or simplified, task-driven alternatives. Foundational systems such as the humanoid hand in [5], DLR/HIT Hand II [6], and the DLR Hand-Arm System [7] emphasized high-DoF kinematics and multisensory integration. Later designs like [8], [9], and [10] incorporated biomimetic actuation and compliant skin, targeting contact-rich interaction. Single-print fabrication of monolithic hands enhanced rapid prototyping [11]. However, these designs often lack integrated perception and scalability/reliability for learning. In contrast, our design balances dexterity and simplicity through a tendon-driven, 16-DoF architecture with wide-FOV wrist cameras, engineered specifically for scalable imitation learning.

**Imitation learning and generative control.** End-to-end imitation learning approaches have recently emerged as a promising paradigm for training robot manipulation policies, enabling robots to learn complex skills directly from demonstration data. This surge has been driven by the adoption of generative modeling techniques in robotics, typically applied to large-scale datasets, predominantly involving two-finger gripper platforms. End-to-end autoregressive Vision-Language-Action models (VLAs) [12, 13, 14, 15] pioneered the application of autoregressive transformers for action generation, demonstrating the ability to learn diverse skills with a single model, at the cost of slow inference and suboptimal action tokenization. To benefit from the strengths of autoregressive VLA models for the high-frequency, dexterous manipulation case, tokenization schemes alternative to naive action binning have been proposed (e.g., FAST [16]).

Alternatively to autoregression, learning high-frequency control policies from demonstrations with generative model techniques can be achieved with the use of action chunking. The first of such approaches was the VAE-based ACT [17]. Further development in methods for fast inference at high control frequency leveraged Diffusion Models [18, 19] or Flow Matching [20] over an action chunk, such as Diffusion Policy [2] or PointFlowMatch [21]. $\pi_0$ [22] illustrates that flow matching-based models can be successfully scaled up and combined with Vision-Language Model (VLM) backbones to enable diverse, language instruction-driven robot behaviors.

**Generative control for scalable (humanoid) dexterity.** Several systems have been introduced to enable scalable data collection and policy training for robots of varying dexterity. Universal Manipulation Interface (UMI) [23] introduced a low-cost, portable data collection and policy interface for two-finger gripper robots, agnostic to robotic arms and compatible with static and mobile robot systems [24]. Follow-up works [25] have investigated the data quantity and diversity scaling laws of the UMI Diffusion Policy. Other works [26] have applied the Diffusion Policy model to lower-DoF humanoid dexterous hands. Several works have demonstrated the benefits of pre-training on human video data [27, 28, 29].

## B Robot Setup

### B.1 Robot hand pose retargeting

Agnostic to the hand capture method used for data collection, our method requires re-targeting of human hand poses to robot hand joint angles. To do so, we employ the key-vector-based re-targeting method from [30, 27] (Fig. 4.d), in which the "keyvectors" $v_i^h$ and $v_i^r$ between palms and fingertips for the human and robot hand are used to compute an energy loss function $E\left((\beta_h, \theta_h), q\right) = \sum_{i=1}^{15} \|v_i^h - (c_i \cdot v_i^r)\|$, which minimizes the distance between human hand poses $(\beta, \theta)$ and robot hand poses $q$, with $c_i$ being a scale parameter.

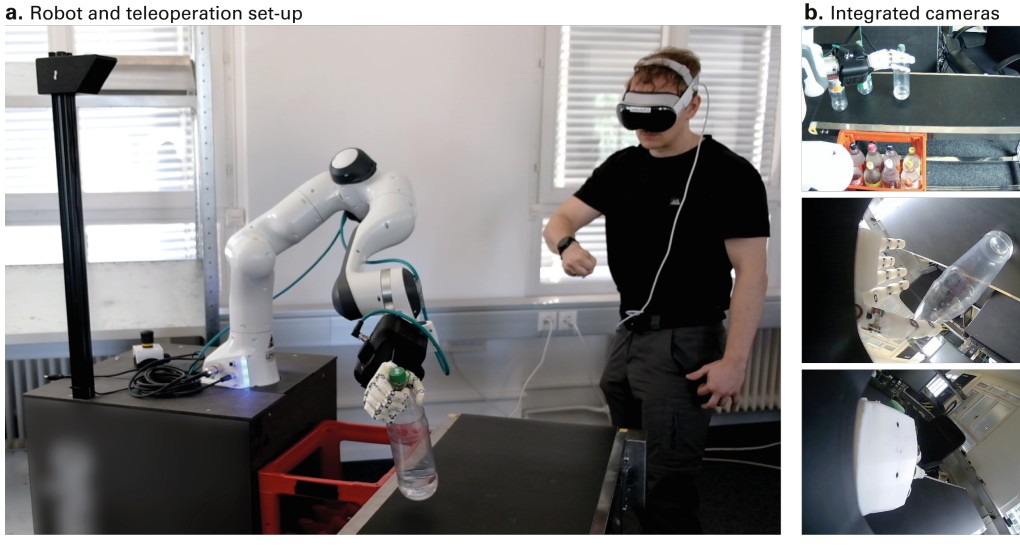

**a.** Robot and teleoperation set-up

**b.** Integrated cameras

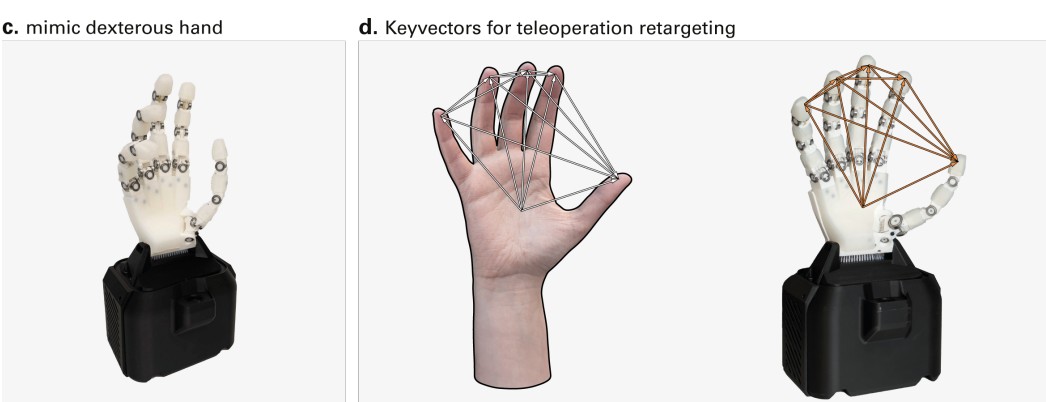

**c.** mimic dexterous hand

**d.** Keyvectors for teleoperation retargeting

Figure 4: System overview and teleoperation setup. (**a**) On the left, the robot station, with a mimic robotic hand mounted on a 7-DoF Franka Emika Panda arm. On the right, a data collector using an Apple Vision Pro for teleoperation. (**b**) Synchronized visual inputs from wrist-mounted fisheye cameras (below and above the wrist) and an external overhead camera. (**c**) mimic hand, featuring 16 DoFs with soft skin contact surfaces and tendon actuation. (**d**) Visualization of keyvector representations used for retargeting human hand poses to robot joint configurations during teleoperation.

## C   Model recipe and Data Collection Protocol

### C.1   Key ingredients for generative control for highly dexterous manipulation

Our recipe relies on several key design choices for state and action representation that significantly boost performance and generalization for dexterous policies:

**Cartesian target end effector pose as action.** Following [23], we use the Cartesian target pose for the end effector as the action, commanded to the low-level impedance controller. This improves robustness to object position changes and promotes arm-agnostic policies. Crucially, the action is the *target* pose from the teleoperator, not the currently observed pose. The policy essentially learns to act as a drop-in replacement for the teleoperator.

**Cartesian proprioceptive end effector pose as state.** The end effector state uses the Cartesian pose derived from robot proprioception.

**"Relative" end effector action representation.** Choosing the right reference frame for Cartesian end effector actions is crucial. Common options include **Absolute** (poses in a global frame), **Delta** (each pose relative to the immediately preceding one), and **Relative** (all poses in the action chunk

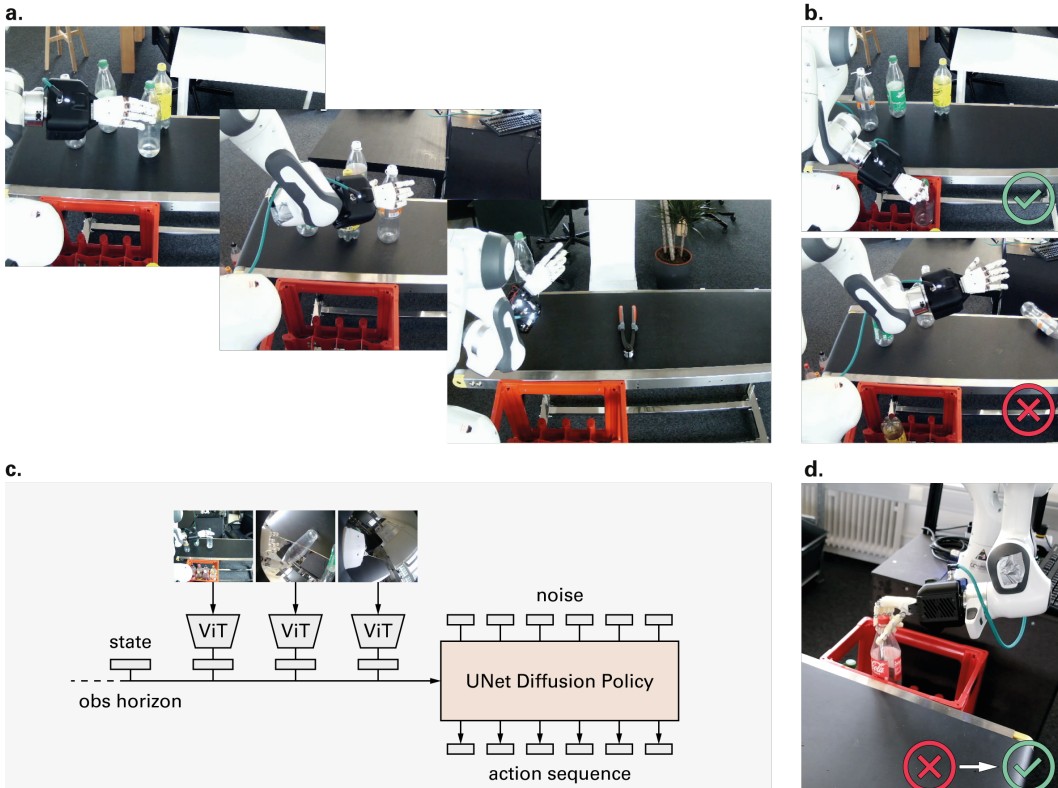

Figure 5: The **mimic-one** data collection protocol and policy recipe. (**a**) Teleoperation data collection. Variation in the setting involve randomize object positions, robot starting pose, "task config", and distractors. (**b**) Data labeling and filtering (removing failures, non-stable grasps and suboptimal completions). (**c**) The diffusion policy model architecture, receiving as conditioning input an observation horizon with encoded RGB images and proprioceptive state inputs, predicting the future action chunk. (**d**) Self-correction trajectory collection based on common failure modes. Robot and workspace are positioned in a failure state, and correction episodes are collected.

relative to a single 'base pose') [17, 23]. For dexterous manipulation requiring adaptation to varying object positions, we find the **Relative** representation superior for generalization. It makes the planned trajectory segment largely independent of the robot's absolute starting position. A critical implementation detail for correctness and stability is selecting the correct base pose: it must be the *last observed proprioceptive end effector pose* (i.e., the final state in the input observation horizon $o_{t-H_o:t}$). Using the first commanded target pose from the current action chunk as the base frame would mismatch the policy's conditioning at inference time, which only includes observed states.

**"Relative" end effector state representation.** Similarly, the end effector state horizon is represented relative to the last observed proprioceptive end effector pose.

**Absolute hand joint angles as action and state.** For the dexterous hand, we use absolute joint angles for both state and actions. This allows the policy to learn meaningful grasp configurations directly, while this information would be lost if using relative angles.

**6D rotations.** All end effector rotations are represented using the 6D upper-triangular rotation representation [31], which is well-suited for neural network learning due to preserving continuity.

## C.2 A data collection protocol to boost success rates

Existing works often focus on either limited, in-distribution demonstrations, or attempt to achieve generalization and high task success rates by simply scaling up data collection to a very high degree. We observe that irrespective of scale, obtaining high success rates for general purpose tasks with

approaches purely based on Imitation Learning is strongly facilitated when following a specific **data collection protocol**, structured as follows:

**(1) Data collection:** Initially, data collection is performed with teleoperation, attempting to perform the entire task successfully in each episode (Fig. 5.a). While doing this, one should adhere to the following: **(1.1)** For each different episode, randomize the starting pose of the robot within the workspace and the position of any movable task object. **(1.2)** Add

Table 1: Dataset statistics.

| Task Name | # Success Eps | # Filtered Eps | # Task Configs |
|---|---|---|---|
| Bread Pick | 1830 | 550 | 5 |
| Bottle Sort | 2420 | 882 | 8 |
| Battery Insertion | 1192 | 528 | 5 |

distractor objects (objects not involved in the task) in the workspace in approximately 50% of the episodes. **(1.3)** We define a "Task config" as the overall setting in which an episode is collected. Specifically, we consider as a different task config a situation in which one of (**a**) table color or type; (**b**) Background; (**c**) Lighting conditions; has changed. Following the observations from [25], we apply the scaling-law optimal ratio of different task configs to the overall number of collected episodes, which means that we change task config approximately every 100 episodes collected.

**(2) Data labeling:** Once the first data collection step is completed, the original data collector must manually label each collected episode as either a success or a failure, depending on whether the task is completed (Fig. 5.b).

**(3) Data curation:** A different person conducts further filtering, eliminating trajectories involving non-stable grasps, erratic motions, and ambiguous task completions.

**(4) Policy training:** A first policy is trained and evaluated in-distribution. Common failure modes are identified and documented.

**(5) Repeat — collect self-correction trajectories:** based on the common failure modes. The robot and workspace are positioned in a failure state before starting the episode collection (Fig. 5.d). After a new batch of self-correction trajectories has been collected, one can begin training again.

Overall, we collect episodes over 3 main task families. We select tasks based on the desire to test for: 1. Overall capability for pick and place in diverse, real-world settings. 2. Completion of dexterous tasks that are not easily achievable with two-finger gripper solutions. 3. Precision in tasks such as assembly and insertion. 4. Illustration of self-correction behaviors.

# D  Training Details

**Observation horizon.** Each training sample consists of an observation horizon $H_o = 2$ and an action chunk of $H_a = 48$ time steps, corresponding to 3.2 s of future prediction at a control rate of 15 Hz.

**Model architecture.** The image encoders are CLIP [3] ViT-B/16 [4] encoders, kept unfrozen at training time. The action denoising UNet uses the architecture from Diffusion Policy [2], with an embedding dimension of 128, kernel size = 5, number of groups = 8, and a training-time number of inference step of 16. The UNet is conditioned with a concatenated embedding from the observation horizon. The visual embeddings from RGB camera observations are taken from the CLS token of the ViT encoder. As for the low-dimensional observation, they are not projected, but are directly concatenated with the visual embeddings, before being used for conditioning the UNet.

**Observation and Action Representations** The representation of the robot arm's end effector Cartesian poses, for both observed states and predicted actions, incorporates specific strategies to enhance learning and generalization. Let $P_k = (T_k, R_k)$ denote a Cartesian pose at time step $k$, where $T_k \in \mathbb{R}^3$ is the translation and $R_k \in SO(3)$ is the rotation matrix. The current time step is $t$.

- **"Relative" End Effector Pose Representation.** A key aspect of our methodology is the use of a "Relative" representation for end effector poses. This means that sequences of poses are expressed relative to a common reference frame, specifically the most recent

observed proprioceptive pose of the end effector. Let $P_t^{obs} = (T_t^{obs}, R_t^{obs})$ be the latest observed proprioceptive end effector pose at the current time $t$. This $P_t^{obs}$ serves as the "base pose".

- **State Representation:** The input observation horizon for the end effector poses, denoted as $\mathcal{O}_{ee} = \{P_{t-H_o}^{obs}, \ldots, P_{t-1}^{obs}, P_t^{obs}\}$, consists of $H_o + 1$ observed poses. Each pose $P_k^{obs}$ in this sequence (for $k \in [t - H_o, t]$) is transformed into a relative pose, $P_k^{state\_rel}$, with respect to the base pose $P_t^{obs}$:

$$P_k^{state\_rel} = (T_k^{state\_rel}, R_k^{state\_rel})$$

where

$$T_k^{state\_rel} = (R_t^{obs})^T (T_k^{obs} - T_t^{obs})$$
$$R_k^{state\_rel} = (R_t^{obs})^T R_k^{obs}$$

The sequence $\{P_{t-H_o}^{state\_rel}, \ldots, P_t^{state\_rel}\}$ (where $P_t^{state\_rel}$ is the identity pose $(0, I)$) forms the state input to the policy concerning the end effector.

- **Action Representation:** The policy predicts an "action chunk," which is a sequence of $H_a$ future target end effector poses. These are also represented relative to the same base pose $P_t^{obs}$. The policy outputs:

$$\mathcal{A}_{pred} = \{P_t^{act\_rel}, P_{t+1}^{act\_rel}, \ldots, P_{t+H_a-1}^{act\_rel}\}$$

Each $P_j^{act\_rel} = (T_j^{act\_rel}, R_j^{act\_rel})$ for $j \in [t, t + H_a - 1]$ is a pose relative to $P_t^{obs}$. To obtain the absolute target pose $P_j^{target} = (T_j^{target}, R_j^{target})$ that is commanded to the low-level robot controller, this relative pose is combined with the base pose $P_t^{obs}$:

$$T_j^{target} = T_t^{obs} + R_t^{obs} T_j^{act\_rel}$$

$$R_j^{target} = R_t^{obs} R_j^{act\_rel}$$

- **6D Rotation Representation** All 3D rotations $R \in SO(3)$ for the end effector poses, whether in the observed states or the predicted actions (i.e., $R_k^{state\_rel}$ and $R_j^{act\_rel}$), are represented using a continuous 6D format as proposed by [31]. If a rotation matrix $R$ is given by its column vectors:

$$R = [\mathbf{r}_1, \mathbf{r}_2, \mathbf{r}_3], \quad \text{where } \mathbf{r}_i \in \mathbb{R}^3$$

The 6D representation, $\mathbf{R}_{6D} \in \mathbb{R}^6$, is formed by concatenating the first two column vectors:

$$\mathbf{R}_{6D} = \begin{bmatrix} \mathbf{r}_1 \\ \mathbf{r}_2 \end{bmatrix}$$

This representation avoids discontinuities and gimbal lock issues associated with other rotation parametrizations like Euler angles or quaternions when used directly in neural network outputs.

**Data augmentation.** We apply random cropping with a 0.95 ratio to the RGB images, together with a color jitter random filter. No augmentations are applied to proprioceptive inputs or actions.

**Normalization.** All proprioceptive inputs and actions are normalized to fit within limits of +1 and -1, using statistics computed over the training set. RGB images are normalized according to the ImageNet normalization used for the ViT encoder.

**Training.** Policies are trained with the standard denoising diffusion loss (mean squared error between denoised and ground truth action chunk), using the AdamW optimizer with a learning rate of $3 \times 10^{-4}$, weight decay of $1 \times 10^{-6}$, a cosine decay schedule and a warmup of 2000 steps. Training runs for 120 epochs over the task dataset, which has been observed to be enough for all individual test tasks.

**Inference.** During inference, the policy runs in real-time at 15Hz, predicting new action chunks with a 16-step DDIM sampler. We use a dynamic action scheduler, playing back actions when they are ready, according to their scheduled timestamp. We re-compute the action chunk before its complete execution, after the execution of the 10th action of 48.

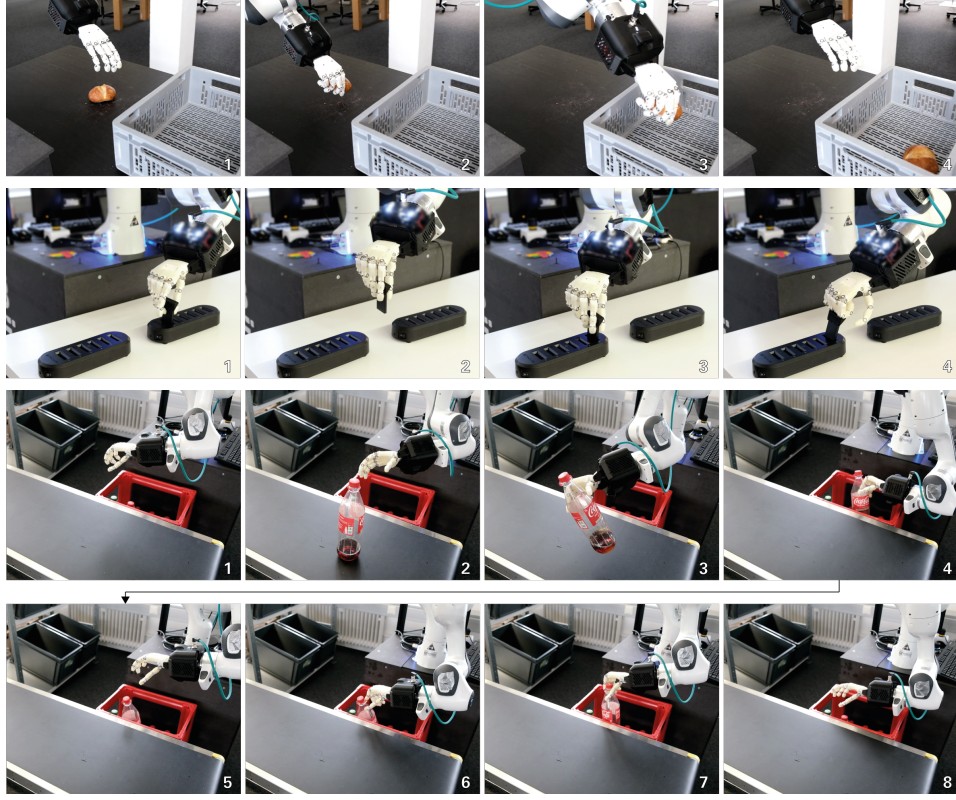

Figure 6: Representative rollout sequences across three benchmark tasks. The mimic-one policy demonstrates smooth, self-correcting behaviors over **bread pick-and-place**, **battery insertion** and **bottle sorting**. We thoroughly illustrate self-correcting behavior for the bottle sorting task: the hand re-orients the bottle post-grasp to achieve successful insertion.

## E   Task Success Criteria

We define success criteria for each task as follows:

- **Bread Pick-and-Place:** The bread loaf must be fully inside the target container and not crushed or dropped during placement.
- **Bottle Sorting:** The bottle must be inserted into a free slot in the rack and remain upright. Corrections after misplacement are permitted.
- **Battery Insertion:** The battery must be inserted into the designated slot and visually confirmed to be flush with the rack (fully punched in).

## F   Failure Mode Taxonomy

To facilitate targeted self-correction collection, we categorize failure modes as follows:

1. **Unstable grasp:** Object is grasped but not securely held; results in drop or slippage during placement.
2. **No grasp:** Missed grasp attempts due to poor object localization or approach trajectory.
3. **Misalignment:** Object is correctly picked but misaligned for target insertion or placement.

These failure modes can happen either during data collection itself, or during policy inference. During data collection, the protocol is to stop the recording of the current episode, and start a new recording involving performing a correction of the current error. If happening during inference, a

descriptive annotation of the error is made and used to create custom "reset scenes" during the next round of data collection, recording custom recovery demonstrations.

## G   Limitations

While mimic-one demonstrates significant progress in real-world dexterous manipulation, several limitations should be acknowledged:

- **Dependence on imitation learning:** The core approach relies on imitation learning, which inherently limits the policy's capabilities to the quality, diversity, and optimality of the human demonstrations. Performance is fundamentally upper-bounded by the teleoperator's skill and the fidelity of the teleoperation system.

- **Data collection scalability and cost:** Generating the required demonstration data, including the crucial self-correction trajectories, necessitates access to the specific robot hardware (mimic hand, Franka arm, cameras) and human operator time for teleoperation and data curation.

- **Single-task focus in evaluation:** Although the proposed recipe is suitable for multi-task learning, the current empirical validation focuses on training and evaluating policies for individual task families. Training separate policies may also be less data-efficient than multi-task learning frameworks that promote representation sharing.

- **Hardware specificity:** The presented policies are trained specifically for the 16-DoF mimic hand and its sensor configuration (wrist cameras, proprioception). Transferring these policies directly to different robotic hands, would likely require retraining or domain adaptation. The system currently lacks tactile sensing, which could further enhance robustness and enable more contact-rich manipulation tasks.

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
