# OpenReview forum: "mimic-one: a Scalable Model Recipe for General Purpose Robot Dexterity"
_robot-learning.org/CoRL/2025/Workshop/Dexterous_Manipulation — CoRL 2025 Workshop Dexterous Manipulation Spotlight_

### Official Review · Reviewer_sZnN · 2025-09-07
**Accept**

**Rating:** 7
**Confidence:** 5

**Review:**

The paper presents Mimic-One, which includes a new 16-DoF tendon-driven hand, a data collection system, and a policy training framework. The paper studies different design choices for state and action representation and formulates a data collection protocol. Combining both, the authors improve policy performance and demonstrate self-correcting behaviors.

Strengths:

1. The paper studies an important problem in robot learning: how to improve dexterous hand manipulation policy performance in the real world.
2. The findings on different design choices and the data collection protocol are useful for the research community.
3. The new dexterous hand hardware will be appreciated if it can be fully open-sourced in the future.

Weaknesses:

1. It would be beneficial to discuss the hardware design principles and how different design choices affect the manipulation policy or data collection process.
2. Some of the design choices and findings have been studied and reported in prior works, though the additional data points are still helpful.

---

### Official Review · Reviewer_z7aF · 2025-09-12

**Rating:** 6
**Confidence:** 5

**Review:**

In Mimic-One, the authors built a framework that includes the following:
* A custom tendon-driven soft hand.
* A corresponding teleoperation system. (I believe retargeting is done under the assumption that motor and joint angles are linearly related.)
* A learning framework that utilizes diffusion policies.
* A dataset of failure cases.
Other than the first item, none of these components are new, so I would encourage the authors to frame them accordingly rather than as novel contributions. That said, I find it valuable that they integrate these elements and demonstrate policies with correction capabilities on soft robotic hands, while also providing insights into how well the components work together.

---

### Decision · Program_Chairs · 2025-09-18

Accept (Spotlight)